# A Food for All Seasons: Stability of Food Preferences in Gorillas across Testing Methods and Seasons

**DOI:** 10.3390/ani12060685

**Published:** 2022-03-09

**Authors:** Jennifer Vonk, Jordyn Truax, Molly C. McGuire

**Affiliations:** 1Department of Psychology, Oakland University, Rochester, MI 48309, USA; jptruax@oakland.edu; 2Zoo Miami, Miami, FL 33177, USA; molly.mcguire@miamidade.gov

**Keywords:** gorilla, food preferences, touchscreen, stability, paired choice, validation, zoo

## Abstract

**Simple Summary:**

We assessed food preferences in three adult male zoo-housed gorillas over a period of 20 months. We used paired-choice tasks on a computer touchscreen and we also presented food pairings by hand using PVC tubes and compared the stability of preferences across methods and seasons. In addition, we assessed whether preferences were stable across offerings within a session. To further validate the use of images on touchscreens to assess food preferences, we showed that these gorillas chose images of highly preferred food items when these images were paired with images of their less preferred foods. These tests can help us understand the stability of animals’ preferences and whether preferences are dependent on the larger test context.

**Abstract:**

Decisions about which foods to use during training and enrichment for captive animals may be based on invalid assumptions about individuals’ preferences. It is important to assess the stability of food preferences given that one-time preferences are often used to inform which items are offered over a longer period of time. Presenting preference assessments using images of food items allows control over factors such as size, scent, and inadvertent cueing but requires validation. We presented three male gorillas with choices between randomly selected pairs of actual food items from their morning meal using PVC feeders. We also presented the gorillas with two-alternative forced-choice tests between images of these foods on a touchscreen computer. Ranked preferences were correlated across method and seasons. Furthermore, gorillas selected images of preferred over less preferred foods in a validation task on the touchscreen. However, selections of some food items changed within sessions, suggesting that preference may be relative to other contextual factors. Researchers should assess how choices affect subsequent preferences to understand whether animals demonstrate absolute preferences for particular food items, or prefer variety.

## 1. Introduction

Food is perhaps the most important resource provided to captive animals and is frequently used as a reward during training and as enrichment. Using preferred food items is essential to ensure that animals receive adequate nutrition, and that they are motivated to perform required tasks and to interact with their environment [1]. Assessing the capacity of different reinforcers to motivate subjects can improve performance in husbandry and research tasks and enhance animal welfare. For example, use of preferred reinforcers can improve the efficacy of training techniques [2,3], reducing discomfort when training animals, for example, to receive injections or place a limb in a restraint [1]. Knowledge of preferences will also be crucial when attempting to stimulate appetite in animals upon recovery from surgeries or during illness. In addition, as pointed out by others [1], identifying highly preferred foods may reduce the need for food adjustment schedules to motivate animals. However, assessing the efficacy of reinforcers requires more than a one-time assessment of ranked preferences. It will necessitate an evaluation of how preferences change over time, and relative to the presentation of other options. Furthermore, past research has identified that trainer and caregiver perceptions do not predict which food items function as effective reinforcers in both humans and animals [2,3,4,5,6]. Zoo personnel may also tend to assume similar preferences for all members of the same species, rather than appreciating individual preferences [5]. Thus, we present validation of an empirical task to assess relative food preferences in nonhuman subjects and a test of the stability of assessed preferences.

Captivity presents the opportunity to investigate food preferences without the confounds of resource scarcity and competition [7]. Food preferences in captivity have been assessed variously using forced-choice tasks, e.g., [7,8,9,10], giving up density measures, e.g., [11,12,13,14], progressive ratio techniques, e.g., [1,10,15,16], multiple-stimulus without replacement (MSWO) procedures, e.g., [1,17,18] and other behavioral measures. Touchscreens are an innovative method to present animals with images of food items that they can select to indicate which food they would like to consume [10,19,20,21]. The use of images presented on a touchscreen has the advantage that it controls for differences in quality indicated by scent, size and other factors that may be difficult to control using actual food items. Additionally, it avoids the potential of biasing the response by holding one item closer to the subject [10], or directing more eye contact to a particular item, for example. It also facilitates randomization of pairings and presentation order of stimuli and allows precise measurement of latencies to make a response. Choices across pairs of items can be ranked to indicate relative preferences for various items. Of course, presentation of food items via images on a touchscreen has weaknesses that are directly related to its strengths; subjects cannot use olfactory or tactile cues, which are important in identifying and assessing real food items and can offset deficits in vision [22]. However, it is important to control the use of such cues so that only information about the type or category of food itself, rather than the quality of a single instance, is used to guide responses. Typically, researchers have assessed preferences for a small number of food items using touchscreen methods, allowing them to present all possible pairs systematically, controlling for side, trial order, and other potential confounds (although one set of researchers [20] presented 38 items to their macaques, *Macaca silenus*). At least one other study [23] presented random pairings of a large number of real food items (66 unique pairs) to captive meerkats, (*Suricata suricatta*). Here, we presented three captive male western lowland gorillas (*Gorilla gorilla gorilla*) with many different food items from their daily morning meal (approximately 40 food items) over a relatively long period of time (20 months) using both manual (real foods) and touchscreen (images of foods) methods, allowing us to validate the touchscreen procedure and to test the stability of food preferences across seasons.

Although touchscreen methods for assessing preferences have the advantage of allowing for the presentation of items that may not be available at the time of testing [10,24], they require validation to demonstrate that subjects perceive the images as representing the actual items depicted. Tests have revealed mixed evidence for picture-object correspondence in various species including primates [25]. For example, gorillas and baboons seemed to confuse photos of food with actual foods because they occasionally tried to consume photos of bananas [26]. Validation of the use of food images can be achieved using a variety of methods. For example, researchers [27] showed that gorillas, chimpanzees, and Japanese macaques chose differently when specific food pairings were presented in blocks, allowing the primates to predict what would be offered next, compared to when the next pairing could not be predicted. When the pairing included one known item and a ‘chance’ symbol that represented a reward that could not be known at the time of choosing, the primates were more likely to choose the ‘chance’ symbol when it was presented with a less preferred food image compared to when it was paired with a highly preferred item. These findings suggest that the primates understood the offerings presented via images on a touchscreen. Other researchers [10] used a different method to validate the use of images on a touchscreen to assess food preferences. They showed that a single gorilla worked harder to access highly preferred versus less preferred foods by touching the pictures of highly preferred foods more often to earn that food. Here, we assessed picture object correspondence by comparing food preferences demonstrated through two alternative forced-choice tasks presented on the touchscreen to ranked preferences obtained via choices between pairs of the same actual foods offered by hand. In addition, we validated the use of pictures of foods by assessing whether the gorillas made categorically distinct responses to foods from the categories “preferred” and “less preferred”. That is, we asked whether the gorillas consistently preferred to select images of foods that they had previously frequently chosen in the food preference tests when they were paired with foods that they had seldom chosen. Preferences to touch images of preferred foods despite non-differential reinforcement would validate our classification of these foods into preferred and less preferred categories.

Previous studies had shown that animal subjects respond to photographs of food that correspond to preferences for actual food items either by selecting images of greater quantities of food items or images of more highly chosen foods when paired with images of smaller quantities of food items or less highly chosen foods. A black bear that demonstrated picture object correspondence [28] spontaneously preferred to select three pieces over one piece of almond depicted pictorially [29] although the gorillas tested in the current study did not show the same preference for images of three biscuits over one biscuit in a previously published study [30]. However, two sloth bears (*Melursus ursinus*) chose high-preference items in 80% of trials in which they were presented with a lower-preference food item and they transferred this preference to novel images of the foods [21]. Lion-tailed macaques (*Macaca silenus*) also learned to select images of foods that they preferred over foods that they preferred less when they received the same food that they selected on the touchscreen. Three of the five macaques transferred this preference to novel images of different high- and low-preference foods, demonstrating that they recognized the images as representing the corresponding foods. Moreover, they chose novel images of foods of medium-preference foods when these were paired with images of low-preference, but not high-preference foods, further indicating picture object correspondence [20]. Based on the previous studies with species more distantly related to humans and the conflicting results with gorillas presented with food images previously [26,30], we were interested in whether gorillas would show similar preferences for images of the preferred foods over less preferred foods. If the gorillas showed a preference for touching images of foods from the preferred category under non-differential reinforcement, it would constitute some evidence that the food images were meaningful to them and may be seen as representative of actual food items that varied in terms of their preferences.

Given that researchers and zoo personnel sometimes operate under the implicit but untested assumption that food preferences are stable, we were also interested in assessing the stability of preferences across different seasons and to compare preferences across winter seasons one year apart. Examining food preferences as a function of season may be important, even in captive settings, as the availability of food items changes with changing seasons and species may experience different energy expenditures as a function of breeding seasons, and hibernation, for example. Demonstrable stability of preferences for food presented as images on the touchscreen over time would lend further credibility to the assumption that choices of photographic stimuli were meaningful for the gorillas. Moderate levels of stability in preference for five food items have been reported in a group of rhesus macaques (*Macaca mulatta*) over a period of six months to one year using a MSWO procedure [1]. A previous study had shown some instability in preference in orangutans (*Pongo* sp.) also with a set of five food items over a shorter period of time (6 months) [17], although black bears (*Ursus americanus*) showed stability in preference of many food items over a year [31]. One might expect greater instability in preference in black bears compared to primates given that hibernating bears may display seasonal variability in prioritization of high caloric intake. Thus, more studies are needed to better understand the stability of preferences in different species.

Although some previous research has investigated the stability of primate food preferences over time [1,17,31], and relative preferences for specific foods over variety [32,33,34,35,36,37,38], research has not often explicitly examined how preferences might change over repeated offerings within the same time period or experimental session. Rhesus macaques have revealed within-session decrements in responding attributable to satiation for a given reward [39]. These researchers suggested that satiation can be attenuated by presenting a mixture of food items. Of some additional relevance, tests of cognitive dissonance have assessed whether animals are likely to prefer food items that were previously chosen over equally preferred foods in a forced-choice procedure [40,41]. These studies suggest that animals—primates in particular—may adjust their preferences in line with their actual selections when forced to choose between two equally preferred options. In a human sample with developmental disorders, varied items, rather than a single item, served as effective reinforcers for basic tasks (i.e., a bar-pressing activity) as shown by higher rates or longer durations of responding [35,36]. However, individual participants responded in unique ways to variation, highlighting the need to assess the best method to present items of differing preference to individuals [36]. Varying the reinforcer can lead to more frequent bar presses [37], and these findings generalize to more naturalistic situations, such as on-task and academic performance [38]. Studies of endowment effects have also shown that preferences may depend upon which foods have previously been in an animal’s possession [42]. Gorillas, along with chimpanzees [43,44] and orangutans [45], have shown the standard endowment effect for both high- and low-value foods [44,46]; however, the strength of the effect differs across ape species. In one previous study [44], foods were labelled as high and low value if they were rarely or commonly available to gorillas rather than being based on preference tests. Food rewards used in experimental tasks are often described as “highly preferred” in the absence of explicit assessments of preference or the stability of those preferences. However, keepers and caregivers are not always accurate when assessing which foods are most likely to motivate animals or humans to perform tasks [2,4]. It is important to consider whether commonly offered foods equate to those that are preferred and therefore to consider how novelty and familiarity affect preferences. Standardized preference test assessments would help researchers determine which foods are actually valued rather than using a potentially arbitrary metric. To assess preferences over repeated offerings, we assessed how these gorillas’ likelihood of choosing frequently presented food items changed across offerings of those food items within sessions.

In the wild, western lowland gorillas eat an extremely eclectic diet, making them excellent subjects with which to explore stability of food preferences [47]. Captive gorilla food preferences have previously been assessed in comparison to chimpanzee preferences [7]. Somewhat surprisingly because of differing body size and diets in the wild, both ape species preferred foods with high sugar to fiber ratios. Given the relatively small numbers of primates of a given species housed at the same facility, cross-study comparisons are essential but will benefit from the use of a standardized procedure. Here, we analyzed whether preferences for foods were consistent when real food items were presented by hand or images of the same foods were presented on a touchscreen and whether these preferences were consistent over four seasons. We examined whether gorillas were more likely to select images of preferred over non-preferred food items even when not differentially reinforced for doing so. We also assessed whether preferences were consistent over multiple offerings within a single session. We predicted that food preferences would be largely consistent within gorillas and across seasons but that preferences within a session may vary if gorillas prefer variety over receiving the same food item over multiple trials.

## 2. Methods

### 2.1. Subjects

Three male western lowland gorillas (*Gorilla gorilla gorilla*) between the ages of 15 and 17 during testing—Chipua (Chip), Pendeke (Pende) and Kongo-Mbeli (Kongo)—participated in these tests between June 2013 and January 2015. The gorillas were half-brothers living in intermittent contact (together most of the time but separated overnight several times a week and for an hour in the mornings for feeding) as a bachelor group at the Detroit Zoo in Royal Oak, MI, USA.

Testing took place in a restricted access indoor area, with each gorilla tested in an individual holding area where they spent approximately an hour every morning for feeding and training while their primary habitats were being serviced. During this period, the gorillas also participated in other tasks including various two alternative forced-choice tasks [48], matching-to-sample training [49], cognitive bias tests [30,50,51], and a conditional discrimination task [24] all presented on a touchscreen.

### 2.2. Materials

The touchscreen tests were programmed in REALBasic and presented on a Panasonic Toughbook Laptop CF19 projected to a 19” VarTech Armorall, Baton Rouge, LA, USA, capacitive touch-screen monitor. The touchscreen monitor was welded to the outside of a rolling LCD panel cart encased within a wooden protective covering. A 1.2 m by 1.2 m plywood ramp was placed under the cart, keeping it level with the front of each gorilla’s indoor holding area so that they could touch the screen through the steel mesh. The laptop was placed on the enclosed shelf behind the touchscreen but accessible to the researcher (JV or MM) who stood behind the cart so that her face was not visible to the subject during the task.

Stimuli consisted of non-copyrighted photographs of 46 food items downloaded from various websites (Appendix B). Foods came from the gorillas’ regular morning meal (i.e., various fruits, vegetables, and primate biscuits) and were similar to the foods presented to gorillas in a previous test of food preferences [7]. The gorillas were fed what remained of their morning meal, even if they had declined to participate, immediately after testing. A PVC chute attached to both the left and right sides of the cart allowed the experimenter to deliver the food reward to the gorilla without any direct contact. Two 45.2 cm long PVC feeders were used to offer the food items by hand during the manual tests.

### 2.3. Procedure

The gorillas participated in test sessions on three mornings a week between 07:00 and 09:00 over a period of approximately 20 months with a break in testing while other studies were being conducted (from March to November 2014). On days where they participated in a preference test, each gorilla participated in a single preference test before participating in cognitive tests. A flowchart of the experimental design appears in Figure 1.

#### 2.3.1. Manual Preference Tests

Forced-choice preference tests using pairs of real food items from an assortment of 46 food items were presented via PVC feeders. Approximately 24 sessions per gorilla were presented from June to August 2013 on three mornings a week between the hours of 07:00 and 09:00 prior to the gorillas’ cognitive testing sessions. The researcher (JV or MM) randomly selected two different approximately equally sized items from the gorilla’s morning food tray and placed the items into two separate identical PVC feeders. They then stepped toward the subject, holding the feeders directly in front of themselves straight out from their sides (Figure 2) and keeping their gaze fixed straight ahead to the gorilla’s hands. Once the gorilla had looked at the food items, the researcher spaced the feeders slightly further apart (approximately hip width) and presented them simultaneously against the mesh separating them from the gorilla. The gorilla would then reach or point toward one feeder and the researcher would tip the food from that feeder into their outstretched hand, record the choice, and place the unchosen food aside. The gorillas always indicated a choice. This continued for approximately 6–10 trials depending on how many unique food items were available to present. The gorillas received all remaining food items in their trays, including unchosen items, following the cognitive test sessions for all gorillas (usually approximately 30 min after the preference tests were conducted).

#### 2.3.2. Touchscreen Preference Tests

The gorillas participated in 106 sessions of food preference assessments on the touchscreen. They completed a single preference assessment on a given test day. Sessions were conducted between June 2013 and January 2015 with a break in testing from March to November 2014. For the purpose of analyzing changes in preference across seasons, we divided the sessions up into groups of 19 sessions conducted in June and July (Summer 2013), 29 sessions conducted in September–November (Fall 2013), 40 sessions conducted in December 2013–February 2014 (Winter 2014) and 18 sessions conducted in December 2014–January 2015 (Winter 2015). Sessions were grouped according to seasons, to keep the number of sessions within each season between 20 and 40, and lastly, to ensure comparability between the winter 2014 and 2015 months. This was similar to the approach taken by researchers [31] who demonstrated stability in food preferences in two American black bears across seasons.

The touchscreen was rolled up on a plywood ramp so that it could be pressed right against the mesh so the gorillas could select images by touching the screen through the spaces in the metal mesh of their habitats. The touchscreen was encased in a covered rolling LCD cart so that the researcher could view the choices on a laptop behind the touchscreen that ran the software. The food tray was placed on a shelf directly below the laptop and behind an aluminum shield so that the gorillas could not see which other foods were available at the time of their choice (see Figure 3).

Images of foods were selected for use in the computer program from the foods that were available to feed the gorillas during that session. The computer program selected images from the designated list of food items for that session and repeated all possible pairings of all food items for as many repetitions as specified by the researcher. A unique aspect of our program was that the researcher did not need to limit the foods presented to a predetermined number of expected foods. Furthermore, the number of pairings of each food item with each other could be selected each day depending upon how many pieces of the various foods were available so the number of foods used and the number of trials/pairings varied between sessions. Due to the limited number of pieces of most food types, the researchers were usually able to present only one set of pairings on a given test day and sometimes the sessions were terminated before each food item was paired with each other food item if they ran out of a particular food item. Across all sessions, foods were presented anywhere from 3 to 159 times depending on how frequently the food item was provided. Unlike most paired preference tests, we were not able to systematically pair every food item an equal number of times with each other food item; see also [23].

Two images were randomly paired and displayed during a trial as seen in Figure 3 in random order. The side location of a particular image was selected randomly within the session. The gorilla could touch only a single image per trial. When the gorilla chose a stimulus, the computer produced a beep, the images disappeared from the screen, and the experimenter rolled a small food reward corresponding to the chosen image down the PVC chute on the side of the cart into the gorilla’s hand. The next trial was displayed after a 500 ms delay. Sessions were ended when there were no more food items available of each type selected for the session or until the selected number of pairings was exhausted. Therefore, the sessions did not always systematically include trials of all possible pairings of food items used that day.

#### 2.3.3. Food Preference Validation

Because rankings of foods from both methods were reasonably well correlated, these data were used to create sets of preferred and less preferred food images for each gorilla based on their own individual preferences. Items were selected from the ranked order of preference from touchscreen preference tests conducted between June 2013 and February 2014. They had to be selected at least 55% of the time to be included as preferred items and less than 45% of the time to be included as less preferred items; thus, slightly different numbers of food items were used for each gorilla. For Chip, seven foods were used in the preferred food category; tomatoes, turnips, broccoli, watermelon, Brussel sprouts, beets, and cauliflower. Similarly, seven food items were used in the less preferred category; orange, grapes, green beans, bananas, parsnips, radish, and lettuce. Each of the foods were presented three times within a session, except for lettuce and cauliflower, which were presented twice, to create 20 pairings/trials for a session as dictated by the software. For Kongo, the following seven foods were used as preferred—Brussel sprouts, biscuits, cauliflower, tomatoes, turnips, watermelon, and radish; and less preferred—green beans, cucumbers, pineapple, celery, parsnips, orange and bananas. Radish and orange were presented twice; all other foods were presented three times. For Pende, there were nine preferred items; parsnips, tomatoes, beets, watermelon, radish, bell peppers, orange, turnips and biscuits. All of the foods were presented twice except for tomatoes and parsnips, which were presented three times. For Pende’s less preferred items, there were seven items; grapes, green beans, bananas, pineapple, lettuce, pear and yam. Each were presented three times except for pear, which was presented twice. Only a single image was used to represent each food item and the same image was used in every trial in which that food item was presented.

Stimuli from each of the two categories were randomly paired and presented in random order. On each trial, one image from each category appeared with side counterbalanced (left or right) within category types within a session. Gorillas were rewarded regardless of their choice. That is, when the gorilla touched either image, he heard a tone and received a (potentially unrelated) food reward from the PVC chute attached to the cart. We could not guarantee that a chosen food would be available in the food tray for this test, so foods were selected randomly for this task. The next trial began after 500 ms. Each gorilla completed 20-trial sessions until they chose the preferred foods on 16/20 trials (80%) for four consecutive sessions (Chip, Kongo) or for 24 maximum sessions (Pende). We established this cut-off for the number of sessions because we wished to capture untrained preferences and we did not want to undermine the association between choosing an image and receiving the food depicted. These sessions were conducted in April 2014.

### 2.4. Statistical Analyses

Only eight foods were available across all four testing time periods and presented at least four times during each of these seasons and during the manual food choice tests; apples, beets, broccoli, carrots, cauliflower, tomatoes, biscuits, and Romaine lettuce. The latter two food items were offered to the gorillas every day. The proportion of times that each of these eight foods was chosen out of all the times it was presented to each gorilla was used in the statistical analyses to assess stability of preferences across methods and season. Although an imperfect measure because each food was paired with a random selection of other foods throughout testing (not all of which were included in analyses due to the small number of trials in which they were involved) and for differing numbers of trials, using the proportion of trials in which the food was chosen standardized the measure to some extent. This is the same approach taken by other researchers [23] in their study of group-housed meerkats and by another research group [5] in their multi-species study. It should be noted that our data involved dependencies and pairs were not presented systematically to allow for the usual method of assessing preferences from a paired-choice procedure. Thus, the results of our statistical analyses should be qualified with attention to these limitations.

To compare preferences across methods, Wilcoxon signed ranks tests were used to compare the ranked order of preference for the eight foods in the manual tests to the touchscreen tests conducted during the same season (Summer 2013).

To calculate stability for ranked preferences of the eight food items that were presented consistently across seasons, Kendall’s coefficient of concordance, which ranges from 0 (no agreement) to 1 (perfect agreement), was used as in [1]. In addition, to assess how preferences for specific food items might vary by season, we used generalized linear model (GLM) analyses of the proportion of choices of each food with a gamma distribution using a log link function. We used GLMs to control for subject as a random factor and to account for the nested structure of the data; foods were nested within seasons. Proportion of choices of each of the foods included in the analyses (see below) was distributed normally and displayed no skew (based on visual inspection of a histogram).

To calculate stability of preference for each food item depending on the order of presentation of that food item within sessions, we selected ten foods for inclusion that were presented a minimum of 37 times for any given subject. Most of the ten selected foods were presented more than 100 times for each gorilla across all testing sessions. We counted the number of trials in which a food item was selected on its first, second, third, fourth, or fifth presentation within a session, and calculated the proportion of times chosen out of all the times that food item was presented for each order position. We conducted GLM analyses of the proportion of choices of each food with a gamma distribution using a log link function. We used GLMs to account for subjects as a random factor and the nested structure of the data; trials were nested within sessions, which were nested within food types. We treated order position and food item and their interaction as fixed factors. Proportion of choices of each of the foods included in the analyses (see below) was distributed normally and displayed no skew (based on visual inspection of a histogram). It should be noted that there was some dependency between choices of the food types since some of these foods were paired with each other across trials. Based on the small number of presentations of approximately 30 presented food items that were not included in analyses, we could not analyze the trial by trial data. Planned simple contrasts were used to assess which differences were significant based on reported main effects. No corrections for multiple comparisons were applied because these tests were used to describe main effects rather than to explore unique hypotheses.

All statistical analyses were conducted using SPSS v. 28.

## 3. Results

### 3.1. Preferences across Methods

There were no significant differences in ranked preferences between touchscreen and manual methods in the same season (Summer 2013), according to Wilcoxon signed ranks tests for each gorilla; Chip, *Z* = −0.07, *p* = 0.943; Kongo, *Z* = −0.17, *p* = 0.865; Pende, *Z* = −0.271 *p* = 0.786.

### 3.2. Preferences across Seasons

The gorillas showed moderate stability in their average proportion of choices of the eight foods (apples, beets, biscuits, broccoli, carrots, cauliflower, romaine lettuce, and tomatoes) that were offered consistently across seasons (Summer 13, Fall 13, Winter 14, and Winter 15; Kendall’s *W*s for Chip = 0.51, for Kongo = 0.70, and for Pende = 0.43). For all gorillas together, Kendall’s *W* = 0.45.

We conducted a mixed generalized linear model with gamma distribution on the average proportion of choices of the same eight foods that were offered consistently across seasons with food and season (SU 13, FA13, WI14, WI15) and their interaction as fixed factors and subject as a random factor. The analysis revealed significant effects of food (*F*_7, 64_ = 15.90, *p* = 0.001) and a significant interaction between food and season (*F*_21, 64_ = 2.48, *p* = 0.003). Simple contrasts comparing proportion of choice of tomatoes to all other foods revealed significant differences between tomatoes and all other foods (all *t*s > 1.96, all *p*s < 0.05, Figure 4). Only proportions of selections of beets (*F* = 5.33, *p* = 0.002) and broccoli (*F* = 3.98, *p* = 0.010) differed significantly across seasons.

### 3.3. Within-Session Changes in Preference

A mixed generalized linear model with gamma distribution on the proportion of choices of the ten most often offered foods (bean, beets, broccoli, Brussel sprouts, carrots, biscuits, cucumber, romaine lettuce, rutabaga, and tomatoes) across the first five trials in which they were offered within touchscreen sessions with food and trial (1–5) as fixed factors and subject as a random factor revealed significant effects of food (*F*_2, 130_ = 3672.00, *p* < 0.001) and trial (*F*_1, 130_ = 33.16, *p* < 0.001) and their interaction (*F*_2, 130_ = 5.72, *p* = 0.004). The preference for tomatoes (the most preferred food) was significantly different from all other foods except for beets (all *ts* > −3.01, all *ps* < 0.003). The preference for beans (*t* = 2.16, *p* = 0.033, 95% CI: 0.00, 0.08), broccoli (*t* = 12.48, *p* < 0.001, 95% CI: 0.12, 0.16), carrots (*t* = 2.65, *p* = 0.009, 95% CI: 0.02, 0.13), cucumbers (*t* = 2.65, *p* = <0.009, 95% CI: 0.03, 0.20), romaine lettuce (*t* = 4.31, *p* < 0.001, 95% CI: 0.04, 0.11), and rutabaga (*t* = 3.22, *p* = 0.002, 95% CI: 0.01, 0.05) changed across trials (Figure 5).

Figure 5 displays the variability in individual preferences. Only data from the first four trials are plotted because of the fewer trials for which foods were offered five times within a session. When only the first four trials were included in the statistical model, food (*F*_9, 98_ = 6.52, *p* < 0.001) and trial (*F*_1, 98_ = 5.07, *p* = 0.027) still significantly predicted choice, but their interaction did not (*F*_9, 98_ = 1.79, *p* = 0.079). In the reduced model, preference for tomatoes differed from all other foods except for beets, Brussel sprouts, and biscuits (all other *ts* > 2.00, *ps* < 0.050). Only preferences for broccoli (*t* = 2.36, *p* = 0.020) and cucumbers (*t* = 3.14, *p* = 0.002) significantly differed across four trials. Preference for these less preferred foods seemed to increase over trials. Model fit was better for the model excluding data from the fifth trials (Aikake Information Criterion, AIC = −90.50 versus 56.54).

### 3.4. Preference Validation

Two of the gorillas (Chip and Kongo) chose preferred foods over non-preferred foods on more than 80% of trials across four consecutive sessions (for Chip, within 19 sessions and for Kongo, within 17 sessions). Chip chose the preferred food on 16/20 trials (80%) and Kongo chose the preferred foods on 14/20 trials (70%) in the first session. All gorillas chose the preferred food images at above chance levels across all sessions according to one-tailed one-sample t-tests comparing the proportion of trials on which preferred foods were chosen to chance, 0.50 (Chip, *t* = 12.42, *p* < 0.001, 95% CI: 4.24, 5.97; Kongo, *t* = 8.67, *p* < 0.001, 95% CI: 3.25, 5.37; Pende, *t* = 5.79, *p* < 0.001, 95% CI: 1.14, 2.42). Figure 6 shows the average number of trials on which they chose the preferred foods across blocks of four sessions.

## 4. Discussion

Our goal was to validate the use of touchscreen presentation of food images to assess stability in food preferences over time. The two other studies with gorillas using touchscreens used only images on the touchscreen [10,19]. The fact that we used both manual and touchscreen methods allowed us to validate the use of photos by comparing choices made using both real and photographic stimuli. Although we were limited to calculating correlations between the ranked preferences for only eight items that were presented several times using both methods during the same season, we did show that the preferences for these foods were correlated across methods. Although eight items represent a small sample of all the foods presented across testing, it is more than the number of foods presented in related studies (although see [20] with macaques). For example, we presented approximately twice as many foods to the gorillas than in one other test of gorilla food preferences [7] and more than the four or six food items used in other previous studies [10,19].

We also showed that ranked preferences for the eight items that were presented across all seasons were moderately correlated when comparing preferences across seasons. These findings suggest that food preferences are accurately assessed using this technology and that food preferences are reasonably stable over time. We further validated this conclusion by presenting the gorillas with pairs of randomly matched food images, with each pair including an image of a preferred item and an image of a less preferred food item. The gorillas were not differentially rewarded for choosing preferred foods and they were not specifically rewarded with the food item they chose (because it was not always available during testing, except for biscuits and lettuce). Two of the gorillas chose preferred foods more than less preferred foods in the first session. That they all chose the preferred items at higher rates than the less preferred items (and at rates greater than predicted by chance) despite not being differentially reinforced for these choices validates our classification of foods as preferred and less preferred based on our paired-choice tests.

Our results add to the literature suggesting that touchscreen images are seen as representing real food items [10,20,21,27,28,29,30]. Primates in previous studies demonstrated transfer of preferences from trained stimuli to novel images of the same food, indicating that they represented the images as depicting particular foods [19]. Even more relevant, three of five lion-tailed macaques [20] spontaneously chose images of their preferred foods when presented with novel images of different foods as a measure of transfer. These subjects also chose these new high-preference food images at higher rates compared to images of medium-preference foods, which, in turn, were chosen more often than images of low-preference foods, further validating the presentation of food images on touchscreens. These authors [20] also used four different images of each food to mitigate against the likelihood that macaques simply associated higher rewards with particular images without recognizing the images as representing the actual foods. Other researchers [21] followed a similar procedure to validate the use of pictorial stimuli to evaluate food preferences in two sloth bears. The bears also spontaneously chose photos of more preferred foods over less preferred foods and transferred this preference to novel images. Taken together, these studies support the use of images of foods presented on touchscreens to assess real food preferences.

Having validated the methodology, we found both consistencies and inconsistencies with previous tests of food preferences in gorillas. Although there were individual differences in preferences across all food items, as with other primates [1,17], there were some preferences common to all three gorillas that may reflect a preference for lower fiber foods. Previous research with gorillas both in captivity [7] and the wild [52] showed that gorillas preferred foods higher in protein and lower in fiber. The gorillas tested here preferred tomatoes over the other foods, especially Romaine lettuce and green beans. Tomatoes are lower in fiber than many of the other fruits and vegetables presented here (Appendix A). A preference for tomatoes was also exhibited by two other samples of gorillas [7,19], four chimpanzees, and seven Japanese macaques [19,27]. It should be noted that photos of tomatoes were presented as a novel food against more familiar food photos in [19], so the preference in that study may partially reflect a preference for novelty. However, the same gorillas showed a strong preference for tomatoes regardless of whether foods were presented in blocked pairs or randomized pairs in a later study by the same authors [27]. One of those gorillas [19] also participated in preference tests using images of four of the same foods used here; grape, carrot, turnip, and cucumber [10]. This gorilla showed a strong preference for grapes, over these other foods using both preference tests and effort to obtain measures. Although the gorillas in the current study did not show a strong preference for grapes, their lower preference for cucumber is comparable, but see [7].

Different preferences among different gorilla troops may partially reflect the fact that apes’ food preferences may be learned socially through observing others’ preferences, as has been shown in bonobos [53]. Other authors [19] also pondered whether similar preferences among their macaques were due to social influence but dismissed this possibility. Preferences in captivity are bound to differ from preferences exhibited by wild primates that are more likely to deal with scarcity and fluctuations in food availability. Chimpanzees’ preferences may be altered by the social context and food scarcity [9], but those factors should not have influenced the current study in which gorillas were routinely separated for feeding and for the preference tests. Contrary to wild mountain gorillas that preferred leaves with high sugar content [52], these gorillas did not show strong preferences for foods high in sugar such as bananas (unlike [7]) or carrots. These gorillas also showed stronger preferences for parsnips, beets and broccoli relative to those tested previously [7], but the preference for beets and broccoli were less stable across seasons than their other preferences.

Beyond simply documenting animals’ food preferences, it is important to have a sense of whether these preferences are stable over time and even within sessions. We presented a larger number of food types over a longer period of time than most food preference studies with primates, e.g., [1,7,9,10]. We showed that ranked preferences for the eight items presented across all seasons and both manual and touchscreen methods were moderately correlated. This finding was consistent with the results of a generalized linear model that showed no main effect of season. This moderate stability is also consistent with a study showing stability in food preferences with a wide variety of food items over a one-year period in black bears [31] and an assessment of choice of five food items over a period of 6 months to one year in macaques [1]. Beets and broccoli were the only foods for which preferences significantly differed across seasons. We had expected that the preference for lettuce and other greens may be higher during the winter months when there were fewer natural foraging opportunities in the outdoor habitats, but this did not seem to be the case. We had otherwise predicted that food preferences would be relatively stable over time, so we do not have an explanation for shifts in preferences for these two foods. We suspect that the observed changes may be an artifact of our method using a random rather than a systematic pairing of food items, which is a limitation of our study.

Despite its limitations, our study is one of the only studies to examine changes in likelihood to choose particular foods within sessions in nonhuman primates. Although imperfect, our within-session analysis suggests that preferences for certain foods may shift after multiple consecutive presentations. Previous studies have shown that both human and nonhuman primates may prefer variety over repeated presentations of highly preferred foods [36,37,38,54], so repeated presentations of higher availability foods, such as biscuits and Romaine lettuce, may have diminished the proportion of trials on which they were chosen given the food monotony effects shown in humans and other primates [32,33,34,35,36,37,38].

Our study differed from previous studies of food preferences in gorillas [7] in several ways. First, we presented both real food items and photos of food items on a touchscreen computer, whereas others [7] presented only real food items. Unlike other tests of food preferences in gorillas, we were able to compare choices using real food items to choices using photographs of the same food items. We used a forced-choice test where the gorillas received only the food they selected, whereas others [7] recorded which of two foods was chosen first in both indoor and outdoor settings. We varied which foods were paired on each trial, whereas others presented the same pair of foods within blocks of up to 90 trials per pairing [19]. Previous studies [10,19,20] provided primates with a training phase in which they associated touching a single food image with receiving that food item. One of these teams [20] also showed the real food item to the macaques as they chose the image on the touchscreen. We did not provide such training here, so early trials using the touchscreen may not have accurately reflected preferences. However, the stability of preferences across sessions mitigates somewhat against that concern. Furthermore, our subjects were not naïve to using a touchscreen and had demonstrated that stimuli presented on a touchscreen were meaningful to them, e.g., by showing preferences for larger quantities [48].

Our study also contained several limitations. First, we had a small sample of gorillas. Second, even though we used over 40 unique foods across trials, we had a relatively small number of food presentations for the majority of our food items because of the limited availability of some foods on test days, which was offset by a greater variety for the gorillas. This meant we could use only ten or fewer foods for calculating stability across methods and sessions. Previous studies were also limited to a small number of test items presented for a small number of pairings but these studies were able to pair all food items exhaustively and equally across trials [2,7,9,10,19,20,21]. We had no control over the foods offered to the gorillas. Given the relatively small amount of some food items offered on a given day, we were not able to systematically pair all food items with all other items to create a balanced distribution of our food items. Therefore, any changes in preference for particular food items within sessions or over time was likely highly contingent on the food items that they were paired with. Choices of one food were not independent of choices of another food. Our measure of preference was therefore not a relative measure of valuation but was a likelihood of choosing that item across all pairings—some of these pairings involved the other most commonly presented items but some included items that were presented fewer than five times within a season and so could not be used in analyses. Our inability to present equivalent numbers of all possible food pairings is the primary limitation of our work. However, restricting our analyses to foods that were presented the most often (i.e., at least 37 times for the within-session analyses) mitigates against this concern to some degree. Because each food item analyzed was paired with several different foods across all trials (i.e., foods were paired randomly rather than systematically), it is less likely that any particular food was consistently paired with higher- or lower-preference foods exclusively. Assessing the total proportion of trials chosen out of all of the trials offered accounted for the noise and variance in the particular food that the target food was paired with to the best of our ability given the manner in which the data were collected. These limitations arose in part because this study was designed to provide information about a range of preferred and less preferred foods for use in a Likert scale training procedure [24]. For that study, we needed a gage of preference for a larger set of items. Preference tests have historically been conducted in service to other experimental paradigms [10] because of the importance of determining relative preferences for foods to be used as motivators [1,2,3] or to test cognitive processes that depend upon relative preferences, such as endowment effects, e.g., [42,43,44,45,46], inequity aversion [53], and cognitive dissonance, e.g., [40,41].

Although we examined changes in preference within sessions, which has not been examined in other food preference studies, it would have been ideal to do so while controlling for specific pairings and presenting them more consistently. Fluctuating likelihoods of choosing particular foods within a session are undoubtedly due at least as much to the foods they were paired with on a given trial as they are to changes over repeated offerings. However, the visual trend to choose less preferred foods more and more preferred foods less over trials within sessions suggests a preference for novelty over specific foods, which has also been shown to some degree in capuchins [32,33,34] and a preference for variety, which has been shown in both human and nonhuman primates [35,36,37,38]. An earlier study [19] examined preferences of the same primates for two novel food items in comparison to their preference for foods used in a previous phase of the study, but, unlike the current study, those researchers did not focus on novelty in terms of repetition of presentation of food items within a given session. We hope that our preliminary findings will encourage others to examine changes in food preferences over repeated presentations and preferences for novelty in a more systematic manner. The random presentation of different foods as was performed here, makes it difficult for subjects to predict the next food pairing [27] and may mitigate against choosing the same food repeatedly. It is possible that the gorillas would have chosen more consistently if we had paired the same two foods repeatedly within blocks. Future studies should investigate the impact of varying the predictability and randomness of food offerings.

The use of pairwise forced-choice tests has also been criticized [55], leading others [10] to modify the task—forcing primates to exert differential effort (via differing numbers of required touches to the screen) for preferred foods. This method allows for a valuation of various foods. It would be useful for future studies of food preferences to implement similar cost measures. Despite the limitations of our study, we think it has value in showing the consistency in preference for foods such as tomatoes and lack of preference for foods such as lettuce over time. Future studies should test preferences for the same food pairings across all four seasons for a period of at least two years.

## 5. Conclusions

These findings suggest that presentation of food items using images presented on touchscreen computers may be a valid way to assess food preferences, at least in primates and other species with similar visual systems. Other work suggests that it is also a valid method for testing food preferences in bears [21,28,29]. We have gone beyond previous work to show that the method can be validated both through comparison to choices of actual food items and preferences for images of preferred food items when paired with randomly selected images of less preferred foods. In addition, we have shown that food preferences are relatively stable over time and seasons, at least for food items that were presented relatively often. However, we have also shown that preferred foods are not chosen 100% of the time and that foods that are not selected in initial trials within a session may be chosen more frequently as the sessions go on. This finding has implications for the use of the same food reward over trials within training or enrichment sessions. Others have shown that foods highly preferred in paired-choice tests elicit higher levels of participation in husbandry and research tasks [1,2,3,5], further emphasizing the importance of understanding preferences, their stability and their variability within individuals [1]. We hope that others will consider potential shifts in preferences when using food items as rewards in training and testing sessions. We suggest that two alternative forced-choice tests presented on a touchscreen can be valid for assessing food preferences, especially in conjunction with validation methods such as those presented here or by others [10,27].

## Figures and Tables

**Figure 1 animals-12-00685-f001:**
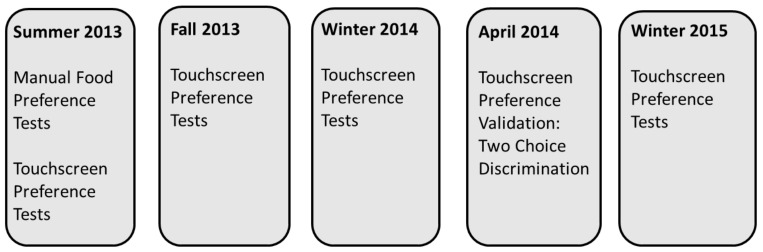
Flowchart of the experimental procedure.

**Figure 2 animals-12-00685-f002:**
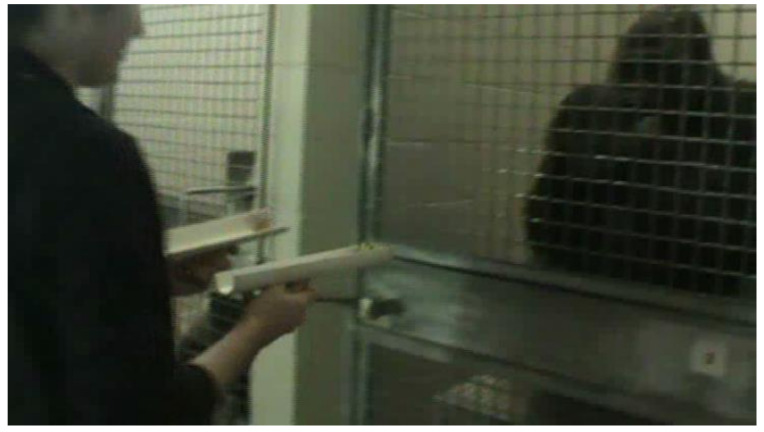
Still from video of manual preference test with Chip.

**Figure 3 animals-12-00685-f003:**
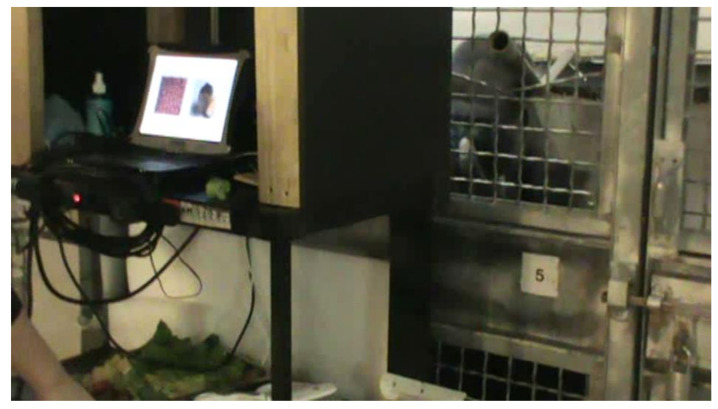
Kongo participating in touchscreen preference test.

**Figure 4 animals-12-00685-f004:**
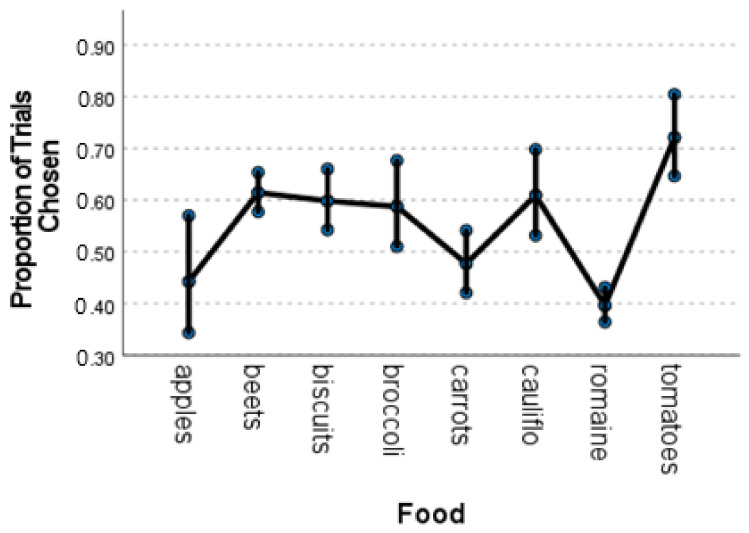
Estimated mean proportion of trials in which foods were chosen by all gorillas using the touchscreen. Error bars depict the standard error of the means.

**Figure 5 animals-12-00685-f005:**
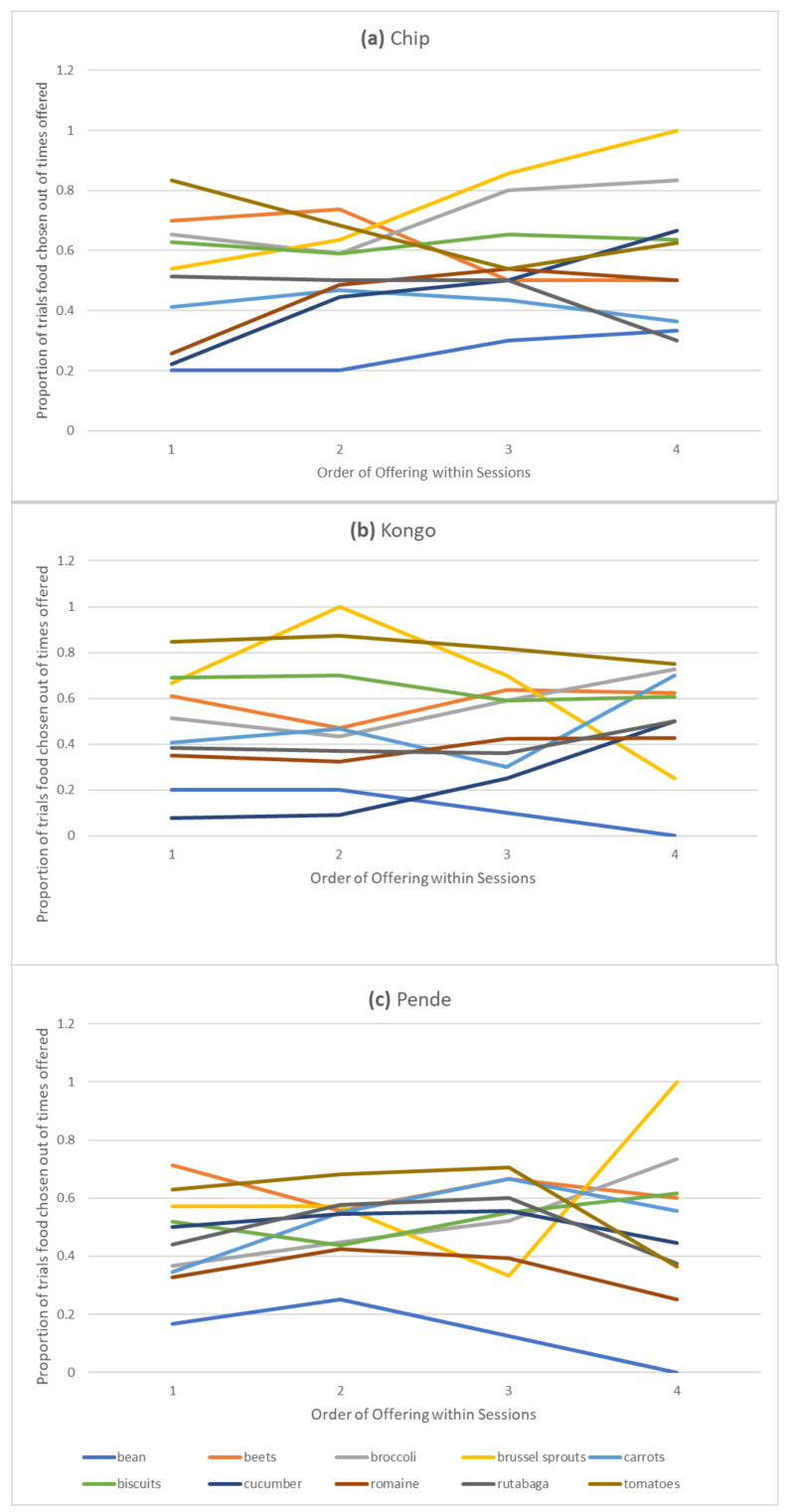
Proportion of choices of each food type across trials in which that food is offered within sessions for each subject.

**Figure 6 animals-12-00685-f006:**
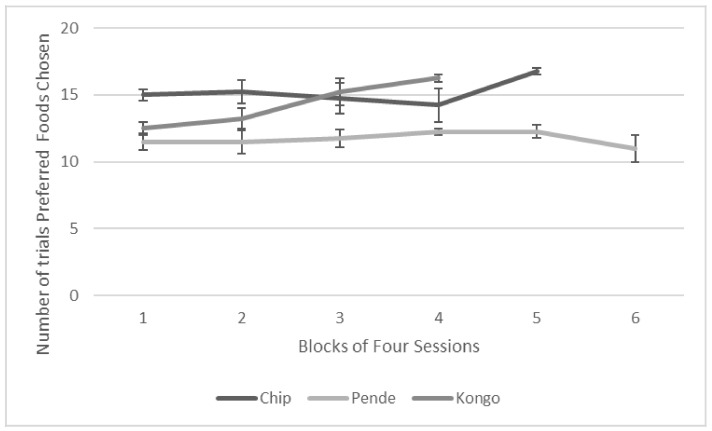
Average number of trials preferred foods were chosen in blocks of four sessions. Error bars represent standard deviations.

## Data Availability

The data presented in this study are available on request from the corresponding author. The data are not publicly available due to the complicated nature of calculating the contingencies from the original data.

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
