# Peer review of "A Food for All Seasons: Stability of Food Preferences in Gorillas across Testing Methods and Seasons"

_animals, 2022, doi:10.3390/ani12060685_

Round 1

Reviewer 1 Report

Thank you for the opportunity to review the manuscript, “A Food for all Seasons: Stability of Food Preferences in Gorillas Across Testing Methods and Seasons.” Empirically determined food preferences can have good utility in captive animal training, enrichment, and management, and I am pleased to see research in this area. The authors’ project adds to the literature in this area by comparing preference assessment methodologies (selection of images on a touch screen compared with choosing from actual food items), preference over a long period of time, and preferences between a variety of foods. I commend the authors on these advances.

However, I do have some concerns about the study. Preference assessment data is based on the percent of trials in which a particular food item is chosen. The food items are paired with other food items in a pairwise manner. The pairwise procedures requires that each food be paired with each other food. As the authors discuss, in some cases (I’m unclear as to how many cases), this did not happen in their study. So, let’s assume that out of 8 foods, the “real” preference is for food A as the 3rd most preferred food and for food B as the 5th most preferred food. If food A happened to only be presented with higher preference foods on that day, it could have a percent of trials chosen of 0%. In contrast, if food B happened to only be presented with lower preference foods, it could be chosen in 100% of trials. Therefore, I’m uncomfortable concluding that changes in preference might be due to season, within-session changes, or anything else other than the variation in the pairings. The project seems to have a large amount of data, and I’d love to see useful analyses come from this work. However, I need more information and more convincing before I’m able to accept the conclusions of the paper given this data limitation. I’m unsure of the best path forward here. Can the authors exclude the sessions in which the pairings were incomplete? I think that would be the best solution if it leaves a usable data set. If not, at minimum, I’d need more information on how many sessions were incomplete. It might also be possible to show that no food was differentially impacted by these omissions. I’m certain the authors have given this more thought than have I, but I remain concerned about all conclusion in the paper given that they are based on the data from incomplete food pairings. I’m hoping the incomplete pairings can be removed. If not, I think the authors need to be more transparent about the number of trials with this issue and do more to justify the validity of their conclusions from these data.

The rest of my comments are secondary to this primary concern and would assume that this concern could be addressed. I will organize them by section:

ABSTRACT:

Please include the number of subjects and sexes.

INTRODUCTION:

Overall, I thought the authors did a nice job of highlighting relevant literature on animal food preference assessments and on the transfer of preference to images. There are quite a few food preference assessments that have been conducted with animals in a variety of settings (farm, lab, companion, zoos). While inclusion of all of these studies is unnecessary, the authors might consider including those previously conducted with primates (e.g., Benz et al., 1992 https://doi.org/10.1037/0735-7036.106.2.142; Fernandez et al., 2004 https://doi.org/10.1207/s15327604jaws0703_2; Harlow & Meyer, 1952 https://doi.org/10.1037/h0058299; Martin et al., 2018 https://doi.org/10.1016/j.beproc.2018.07.002; Polidora & Schneider, 1964 https://doi.org/10.2466/pr0.1964.15.1.55) and/or the studies conducted in zoo settings (e.g., Brox et al., 2021 https://doi.org/10.1002/zoo.21640; Gaalema et al., 2011 https://doi.org/10.1080/10888705.2011.527602; Mehrkam & Dorey 2014 https://doi.org/10.1002/zoo.21151and 2015 https://doi.org/10.1002/zoo.21227). Or, if you choose not to add articles, at least make it clear to the reader what the scope of your review is for the studies you’ve highlighted.

I think it would be helpful to provide more of a justification (i.e., citations, rationale) for why preference might vary by season. It would also be good to provide more of a connection/discussion about whether using high preference foods translates to better welfare, more enrichment use, more effective training sessions, etc.

The overall organization of the introduction section is hard to follow, with results of the studies mixed in with the background literature. I suggest removing all results from the introduction section (e.g., pg.2 line 80, pg 3 lines 103 - 105). Provide the background literature, rationale, a brief description of your plans, and a specific hypothesis for each of your studies. Report the results in the results section only. If needed, you could describe the studies separately (i.e., “Study 1”) and use subheadings.

On pg. 2 lines 81 – 82, more context is needed to set up this comparison/distinction.

How do you make the distinction for preference changing within a session as compared with gorillas simply preferring variability? In humans, variability has been found to be preferred to even to a single high-preference item (e.g., Bowman et al., 1997 https://doi.org/10.1901/jaba.1997.30-451; Hanratty & Hanley, 2021 https://doi.org/10.1002/jaba.835).

I really liked the authors’ focus on availability/novelty and preference (p. 3 lines 130 – 131). It’s a good point, and it might be interesting to explore this further.

Pg. 2 line 85. Please make it clear that this was in a past study in the same location (if that is the case). “Here” is unclear.

Pg. 4 line 160. It is unclear whether you are talking about orangutans or gorillas here. Reword for clarity.

METHOD

In addition to my main methodological concerns previously noted, I have these comments:

Pg. 5 line 201 – Could you provide more context for the variability involved in “sporadically?”

Pg. 5 line 217 – What was the timeframe in which gorillas had to make a choice?

Pg. 7 lines 294 – 297. The phrase “meeting criterion” is a bit confusing here. Provide some explanation/context for why this criterion was chosen and what it signifies. Related, in previous trials, selecting a preferred food resulted in the delivery of that food. If one assumes that delivery of the food reinforced this behavior/selection, then wouldn’t continuing to select these images simply suggest that this behavior persisted under extinction contingencies (as would be expected)? I don’t see what evidence exists that the gorillas are forming categorizations.

RESULTS

Pg. 7 line 304 – Does “these foods” refer to the two presented every day or the longer list?

Pg. 7 line 306 – What does “presented four times” mean here – on four days? Four times within a session? Why was this season chosen?

Pg 7. Line 307. Make it clear that you were comparing preference across the two modalities and that no significant differences in rank were found across the testing modalities.

Pg. 7 lines 311 – 315. Are these preferences based on the physical choice preference assessment? For each assessment, please make it clear what proportion of the tests were physical presentations vs touch screen assessments.

Please justify the use of the Generalized Linear Model with the gamma distribution. You have a very unique data set with only three subjects and a lot of dependency between all data points. Please show how you’ve tested your data to ensure that it meets the statistical assumptions for this test. Also demonstrate how you controlled for family-wise error rates in relation to the post-hoc contrasts.

Figure 3 – Please label as to method (is this also combined?)

DISCUSSION

Overall, the discussion reads well but could probably be shortened and reorganized for clarity. Hopefully the main limitations can be further contextualized by either a reanalysis of the data or the provision of additional information.

Thank you for the opportunity to review this paper. I commend the authors on their work in this area and hope that they can address the methodological/analysis concerns to publish their work.

Author Response

Thank you for the opportunity to review the manuscript, “A Food for all Seasons: Stability of Food Preferences in Gorillas Across Testing Methods and Seasons.” Empirically determined food preferences can have good utility in captive animal training, enrichment, and management, and I am pleased to see research in this area. The authors’ project adds to the literature in this area by comparing preference assessment methodologies (selection of images on a touch screen compared with choosing from actual food items), preference over a long period of time, and preferences between a variety of foods. I commend the authors on these advances.

Thank you very much for these positive comments!

However, I do have some concerns about the study. Preference assessment data is based on the percent of trials in which a particular food item is chosen. The food items are paired with other food items in a pairwise manner. The pairwise procedures requires that each food be paired with each other food. As the authors discuss, in some cases (I’m unclear as to how many cases), this did not happen in their study. So, let’s assume that out of 8 foods, the “real” preference is for food A as the 3rd most preferred food and for food B as the 5th most preferred food. If food A happened to only be presented with higher preference foods on that day, it could have a percent of trials chosen of 0%. In contrast, if food B happened to only be presented with lower preference foods, it could be chosen in 100% of trials. Therefore, I’m uncomfortable concluding that changes in preference might be due to season, within-session changes, or anything else other than the variation in the pairings. The project seems to have a large amount of data, and I’d love to see useful analyses come from this work. However, I need more information and more convincing before I’m able to accept the conclusions of the paper given this data limitation. I’m unsure of the best path forward here. Can the authors exclude the sessions in which the pairings were incomplete?

We understand this reviewer’s concerns. We were transparent about this procedure being a limitation of the study and tried to candidly qualify the results accordingly. However, we do think that restricting our analyses to foods that were presented the most often (i.e., at least 37 times for the within session analyses) mitigates against this concern to some degree. Since each food item analyzed was paired with several different foods across all trials (i.e., foods were paired randomly rather than systematically), it is less likely that any particular food was consistently paired with higher or lower preference foods exclusively. Each food analyzed in the within session analyses was presented a minimum of 37 times with most foods being presented over 100 times. This information is now mentioned in the text. We think that assessing the total proportion of trials chosen out of all of the trials offered accounts for the noise and variance in the particular food that the target food was paired with to the best of our ability given the manner in which the data was collected. This method is akin to using randomization rather than counterbalancing to disrupt effects of potential confounds and is similar to what was used by Brox et al. (2021) who also used random pairings of a large number of food items with meerkats. Although we wholeheartedly agree with the benefits of systematic counterbalancing, we were not able to use that here without severely restricting the food items that could be presented as part of the study because only biscuits and Romaine lettuce were presented regularly in large quantities. Even though foods like carrots and tomatoes were presented fairly frequently, each gorilla might have received only three pieces of tomato or carrot in their food tray that morning. 

The consistency in rankings of food across seasons when they would have been paired with different food items on each session in which they are presented also supports the idea that random ‘luck of the draw’ pairings are not a strong candidate as an explanation for our results. We are not able to merely exclude sessions in which exhaustive pairings of each possible pairing did not occur because such sessions were the norm and we would not have had adequate instances of presentation of any particular food items. Because the exact foods that were used differed on every test day, we would not have enough data using the same food types.

 I think that would be the best solution if it leaves a usable data set. If not, at minimum, I’d need more information on how many sessions were incomplete. It might also be possible to show that no food was differentially impacted by these omissions. I’m certain the authors have given this more thought than have I, but I remain concerned about all conclusion in the paper given that they are based on the data from incomplete food pairings. I’m hoping the incomplete pairings can be removed. If not, I think the authors need to be more transparent about the number of trials with this issue and do more to justify the validity of their conclusions from these data.

The issue really applies to the entire data set as any sessions involving more than a small number of food items did not include exhaustive pairings and we would not be able to compare food choices across seasons or methods if we excluded these sessions. Every test session was unique and involved different items and pairings. We have included more information about this issue and the resulting limitations.

ABSTRACT:

Please include the number of subjects and sexes.

This information already appears on line 23 as well as line 10 of the simple summary.

INTRODUCTION:

Overall, I thought the authors did a nice job of highlighting relevant literature on animal food preference assessments and on the transfer of preference to images. There are quite a few food preference assessments that have been conducted with animals in a variety of settings (farm, lab, companion, zoos). While inclusion of all of these studies is unnecessary, the authors might consider including those previously conducted with primates (e.g., Benz et al., 1992 https://doi.org/10.1037/0735-7036.106.2.142; Fernandez et al., 2004 https://doi.org/10.1207/s15327604jaws0703_2; Harlow & Meyer, 1952 https://doi.org/10.1037/h0058299; Martin et al., 2018 https://doi.org/10.1016/j.beproc.2018.07.002; Polidora & Schneider, 1964 https://doi.org/10.2466/pr0.1964.15.1.55) and/or the studies conducted in zoo settings (e.g., Brox et al., 2021 https://doi.org/10.1002/zoo.21640; Gaalema et al., 2011 https://doi.org/10.1080/10888705.2011.527602; Mehrkam & Dorey 2014 https://doi.org/10.1002/zoo.21151and 2015 https://doi.org/10.1002/zoo.21227). Or, if you choose not to add articles, at least make it clear to the reader what the scope of your review is for the studies you’ve highlighted.

Thank you very much for these positive comments and additional references. We cited several of these publications in our revised manuscript.

I think it would be helpful to provide more of a justification (i.e., citations, rationale) for why preference might vary by season. It would also be good to provide more of a connection/discussion about whether using high preference foods translates to better welfare, more enrichment use, more effective training sessions, etc.

We have added more of this discussion to the very beginning of the introduction and throughout using some of the references kindly recommended by this reviewer.

The overall organization of the introduction section is hard to follow, with results of the studies mixed in with the background literature. I suggest removing all results from the introduction section (e.g., pg.2 line 80, pg 3 lines 103 - 105). Provide the background literature, rationale, a brief description of your plans, and a specific hypothesis for each of your studies. Report the results in the results section only. If needed, you could describe the studies separately (i.e., “Study 1”) and use subheadings.

We included more methods than is typical in the introduction here because the focus of the special issue is on technological advances and how they impact welfare; thus, our rationale for the study largely involved the validation of the method used to assess food preferences. We did remove some sections specifically suggested for removal by the reviewers, however.

On pg. 2 lines 81 – 82, more context is needed to set up this comparison/distinction.

We have expanded our rationale here.

How do you make the distinction for preference changing within a session as compared with gorillas simply preferring variability? In humans, variability has been found to be preferred to even to a single high-preference item (e.g., Bowman et al., 1997 https://doi.org/10.1901/jaba.1997.30-451; Hanratty & Hanley, 2021 https://doi.org/10.1002/jaba.835).

Yes, this is again one of the reasons why we were interested in assessing within session preferences although we acknowledge it needs to be properly examined in a study with systematic pairings. Thank you for these references, which we have cited. We distinguished between seasonal changes and preferences for variety because we have assessments of preference within each session and we tested whether these relative preferences were stable. If these gorillas prefer variety, this should lead to a distribution of choices across foods (which we do see) but should not impact consistency in ranked preferences across seasons.

I really liked the authors’ focus on availability/novelty and preference (p. 3 lines 130 – 131). It’s a good point, and it might be interesting to explore this further.

Thank you. We agree that novelty and availability might have important implications for preferences and should be examined further, which is one of the reasons we wished to examine within session preferences as well as noting which foods were most often available. We just added a sentence here to stress the importance of these factors but we have integrated discussion of these issues throughout.

Pg. 2 line 85. Please make it clear that this was in a past study in the same location (if that is the case). “Here” is unclear.

Thank you for asking for this clarification, which we have made.

Pg. 4 line 160. It is unclear whether you are talking about orangutans or gorillas here. Reword for clarity.

We specified gorillas here.

METHOD

Pg. 5 line 201 – Could you provide more context for the variability involved in “sporadically?”

We have changed to “The gorillas participated in test sessions three mornings a week between 07:00 and 09:00 over a period of approximately 20 months with a break in testing where other studies were being conducted (from March – November, 2014).”

Pg. 5 line 217 – What was the timeframe in which gorillas had to make a choice?

There was no minimum time set for a response. The gorillas always responded immediately.

Pg. 7 lines 294 – 297. The phrase “meeting criterion” is a bit confusing here. Provide some explanation/context for why this criterion was chosen and what it signifies.

We no longer refer to this as a criterion. We changed this to “The next trial began after 500ms. Each gorilla completed 20-trial sessions until they chose the preferred foods on 16/20 trials (80%) for four consecutive sessions (Chip, Kongo) or for 24 maximum sessions (Pende). These sessions were conducted in April 2014.”

We now note, “We established the cut-off for the number of sessions because we were attempting to capture spontaneous preferences and we did not want to undermine the association between choosing an image and receiving the food depicted.”

Related, in previous trials, selecting a preferred food resulted in the delivery of that food. If one assumes that delivery of the food reinforced this behavior/selection, then wouldn’t continuing to select these images simply suggest that this behavior persisted under extinction contingencies (as would be expected)? I don’t see what evidence exists that the gorillas are forming categorizations.

Yes, that is quite plausible that they persisted in choosing the foods they preferred because of the previous training. But the fact that they continued to choose the foods that we had operationalized as “preferred” and didn’t choose the foods that we had defined as “non-preferred” established convergent validity for this operationalization. In this task, we were explicitly pairing foods categorically as highly preferred and less preferred rather than presenting mixed pairings of foods that ranged in preference non-systematically. We agree that we cannot infer categorization without transfer to foods not previously presented but for which we knew the gorillas’ preferences, but we did not have enough foods based on the manual food choice preferences with which to do this. Therefore, we have removed references to categorization of preferred and less preferred food items.

RESULTS

Pg. 7 line 304 – Does “these foods” refer to the two presented every day or the longer list?

We added “eight” here to clarify.

Pg. 7 line 306 – What does “presented four times” mean here – on four days? Four times within a session? Why was this season chosen?

We had a limited number of sessions in which manual preferences were used so this meant four times across all sessions using a given method. So many different foods were presented that very few were presented often. We clarified here.

Summer in 2013 was the only season in which the manual preferences were used and so we wanted to control for season when making the comparison to preferences using touchscreen methods.

Pg 7. Line 307. Make it clear that you were comparing preference across the two modalities and that no significant differences in rank were found across the testing modalities.

We have made this clear.

Pg. 7 lines 311 – 315. Are these preferences based on the physical choice preference assessment? For each assessment, please make it clear what proportion of the tests were physical presentations vs touch screen assessments.

Thank you for noticing this omission. These tests involved only touchscreen data as only this method was used after Summer 2013. We added this information here.

Please justify the use of the Generalized Linear Model with the gamma distribution. You have a very unique data set with only three subjects and a lot of dependency between all data points. Please show how you’ve tested your data to ensure that it meets the statistical assumptions for this test. Also demonstrate how you controlled for family-wise error rates in relation to the post-hoc contrasts.

We examined the distribution of the data using histograms and testing for skew. The data appear normally distributed with no skew. We did not correct for multiple hypothesis testing with the planned contrasts because they were presented to describe and explain main effects, not to test independent hypotheses. We have now included a separate statistical analysis section.

Figure 3 – Please label as to method (is this also combined?)

Thank you for this suggestion as well. We have added “using the touchscreen.”

DISCUSSION

Overall, the discussion reads well but could probably be shortened and reorganized for clarity. Hopefully the main limitations can be further contextualized by either a reanalysis of the data or the provision of additional information.

We have rewritten and reorganized the discussion to reflect the changes made elsewhere and suggestions by all of the reviewers, and to eliminate redundancy.

Thank you for the opportunity to review this paper. I commend the authors on their work in this area and hope that they can address the methodological/analysis concerns to publish their work.

Thank you to this reviewer for their thoughtful critique of our work. We appreciate the concerns with the limitations of our data set and have attempted to address to the best of our ability.

Reviewer 2 Report

In my opinion, this paper addresses an appropriate subject for publication in Animals. However, it requires a substantial revision in order to be publishable. The Introduction is too long, lacks focus, and includes material that should appear elsewhere in the Methods, Discussion or Conclusions. The authors are clearly close to their subject and whilst they know what they are want to convey, they need to recognise that the readers might not be so familiar with the subject and so the authors need to be more precise in their explanations.

I detail my concerns below.

Simple summary: "gorillas" should be "these gorillas". One cannot generalise from three individuals to "all" gorillas, which this sentence implies.

Abstract: this conclusion "... suggesting ... selected recently." might also be due to changing needs of the gorillas.

Abstract: same comment as previous; the authors seem to have a fixed, binary view that doesn't include any changing needs that the gorillas might have.

Introduction: This is too long, and a substantial part of its content belongs more properly to the Discussion. A good Introduction should: specify the scope of an issue; state what is known and not known; identify the priorities; and finally state the objectives of the investigation. This Introduction includes far too much that is extraneous and hence it lacks clear focus.

I am also of the opinion that the multiplicity of use in previous publications of preferred/non-preferred leads to confusion. This is not the fault of the authors; I think that it would be good if they could add some clarity in this paper. For example, it seems to me that prior work sometimes uses the term “preferred” when what is really meant is "readily accessible". Perhaps some critical evaluation might be considered in the Discussion.

Lines 35-36: does the food have to be "highly" preferred or simply acceptable? Would gorillas (or any other species) in the wild reject all but their highly preferred food? What if the individual’s most preferred food was not good for them (e.g. some humans eat and drink foods that are not to their advantage - such as sugar, fat, nicotine)? I find this a curious turn of phrase (highly preferred) and I think that the authors need to consider radically rephrasing this sentence.

Lines 43-45: Touchscreens also preclude the animal from making a detailed analysis of the available food, which might be important in their identifying the food that matches their nutritional needs at the moment, e.g. ripeness, sweetness, spiciness via touch and smell.

Given that this paper is promoting the use of touchscreens as a methodological paradigm, it is my view that it is important to identify in the Introduction what such a methodology can and cannot tell you.

Line 51: Here and elsewhere, the scientific name is absent; it needs to be included at first mention.

Lines 81-88: This section seems to confound two different types of response: (i) preference for three pieces of almond/biscuit is a quantitative choice; (ii) choosing preferred over non-preferred is categorical.

The authors need to redraft this section.

Lines 103-111: "Having demonstrated ... stability in relative preferences." This belongs in the Methods section.

Lines 114-116: But male gorillas about to mate would have particular temporal energetic requirements.

Lines 137-147: Definitely belongs in the Discussion rather than the Introduction.

Lines 153-159: The authors don't mention the effect of novelty. This might be an important factor because in times of scarcity of preferred foods, awareness of novel/unusual foods would be of benefit to survival.

Lines 163-165: This is a conclusion and has no place in the Introduction.

Methods should include the statistical methods that were employed including the statistical program that was utilised (later, the authors have mentioned, Wilcoxon, Friedman and a mixed generalized linear model). Without such information, subsequent researchers cannot repeat the work. The mark of a good Methods section is whether (or not) they are repeatable.

I would also suggest a more explicit statement about the previous experience of these gorillas using a touch screen. Without such a statement, one might imagine that novelty could have contributed to the outcomes of the investigation.

Line 193: "to gorillas" Do the authors mean these gorillas or any gorillas? The authors need to be more precise in their language.

Lines 201-202: "...tests were conducted sporadically ..." How many tests in total? Was the testing routine randomised both over time and across individuals? These questions are actually answered at Lines 225-. I suggest that the authors either include a cross-reference to this explanation or restructure the text to ensure that readers are not posing the same questions as me; this is just a matter of increasing the readability of the text.

Lines 234-238: this is a repetition of the earlier part of the Methods (Lines 182-188). Restructure to avoid repetition.

Lines 242-243: ", programmed in Real Basic," - repetition. I would also check whether the word “Basic” should be capitalized throughout as it is an acronym (Beginners All-purpose Symbolic Instruction Code).

Lines 313-314: I have decoded that SU=summer; FA=fall; and WI=winter. I am not sure that a reader should need to do this; I suggest that the authors explicitly state the season and year on the first mention.

Line 322 and on other occasions: the use of apostrophes to indicate plurals is grammatically incorrect.

Line 326 (Legend to Fig 3): what do the error bars represent? This needs to be stated in the legend.

Lines 335-338: I presume that the CIs are 95%? This should have been stated in the Methods.

Line 349: The acronym "AIC" is not explained.

Lines 403-404: The authors report that preferences for beets and broccoli changed across seasons: how did the preference actually vary? And what reasons are you proposing (you need to provide some sort of explanation, or testable hypothesis)?

In summary, I think that this paper would have some merit, but only after a major revision.

Author Response

In my opinion, this paper addresses an appropriate subject for publication in Animals. However, it requires a substantial revision in order to be publishable. The Introduction is too long, lacks focus, and includes material that should appear elsewhere in the Methods, Discussion or Conclusions. The authors are clearly close to their subject and whilst they know what they are want to convey, they need to recognize that the readers might not be so familiar with the subject and so the authors need to be more precise in their explanations.

We appreciate these comments and have reorganized and rewritten substantial sections of the introduction and discussion.

Simple summary: "gorillas" should be "these gorillas". One cannot generalise from three individuals to "all" gorillas, which this sentence implies.

Done

Abstract: this conclusion "... suggesting ... selected recently." might also be due to changing needs of the gorillas.

We changed to “other contextual factors.”

 Abstract: same comment as previous; the authors seem to have a fixed, binary view that doesn't include any changing needs that the gorillas might have.

We are not sure exactly what the reviewer refers to with this comment. The study is focused on the idea that preferences may change based on a variety of factors, one of which can certainly be changing needs of the gorillas, but this itself is not a very specific construct and not easily objectively assessed. Our point is that most studies of food preferences do not take any changes into account.

Introduction: This is too long, and a substantial part of its content belongs more properly to the Discussion. A good Introduction should: specify the scope of an issue; state what is known and not known; identify the priorities; and finally state the objectives of the investigation. This Introduction includes far too much that is extraneous and hence it lacks clear focus.

Yes, we agree. We included more methods than is typical in the introduction here because the focus of the special issue is on technological advances and how they impact welfare; thus, our rationale for the study largely involved the validation of the method used. However, we have moved some of the information to the discussion where appropriate.

I am also of the opinion that the multiplicity of use in previous publications of preferred/non-preferred leads to confusion. This is not the fault of the authors; I think that it would be good if they could add some clarity in this paper. For example, it seems to me that prior work sometimes uses the term “preferred” when what is really meant is "readily accessible". Perhaps some critical evaluation might be considered in the Discussion.

We have focused our discussion on papers that directly accessed preferences using similar paired choice or MSWO procedures. We take the reviewer’s point that researchers sometimes refer to “highly preferred” rewards when they are simply using something that an animal will readily eat and that is conveniently available for use. We have included some reference to the need to conduct empirical tests rather than relying on keeper judgements etc.

Lines 35-36: does the food have to be "highly" preferred or simply acceptable?

Would gorillas (or any other species) in the wild reject all but their highly preferred food? What if the individual’s most preferred food was not good for them (e.g. some humans eat and drink foods that are not to their advantage - such as sugar, fat, nicotine)? I find this a curious turn of phrase (highly preferred) and I think that the authors need to consider radically rephrasing this sentence.

We agree with the reviewer’s comment here. We have deleted “highly.” We retained “preferred” because we think preferred foods might be needed to motivate subjects in some husbandry and training tasks. We have added citations to some empirical findings that support this view.

Lines 43-45: Touchscreens also preclude the animal from making a detailed analysis of the available food, which might be important in their identifying the food that matches their nutritional needs at the moment, e.g. ripeness, sweetness, spiciness via touch and smell.

This is also an excellent point that we have discussed in the revision. We have acknowledged that we have eliminated the use of these cues, which are important in making real food choices. But to ensure that only the identity of the food type is factored in, we found it important to control for variances in food quality. We cite other literature that supports the idea that primates view images as representing real food items even in the absence of olfactory and tactile cues.

Given that this paper is promoting the use of touchscreens as a methodological paradigm, it is my view that it is important to identify in the Introduction what such a methodology can and cannot tell you.

We agree. We have tried to be balanced in discussing the limitations of the study.

Line 51: Here and elsewhere, the scientific name is absent; it needs to be included at first mention.

We have added throughout.

Lines 81-88: This section seems to confound two different types of response: (i) preference for three pieces of almond/biscuit is a quantitative choice; (ii) choosing preferred over non-preferred is categorical.

Yes, agreed. The point was that animals can understand the information presented in two-dimensional images to make meaningful choices, whether that is about quantity or identity. We have clarified here.

The authors need to redraft this section.

We have done so.

 Lines 103-111: "Having demonstrated ... stability in relative preferences." This belongs in the Methods section.

We have removed methods info and results from the introduction as suggested by all reviewers.

Lines 114-116: But male gorillas about to mate would have particular temporal energetic requirements.

Yes, but we maintain that the unique demands of hibernation are more likely to predict seasonal differences in diet and preferences in hibernating compared to non-hibernating species.

Lines 137-147: Definitely belongs in the Discussion rather than the Introduction.

Agreed. We have moved this section.

Lines 153-159: The authors don't mention the effect of novelty. This might be an important factor because in times of scarcity of preferred foods, awareness of novel/unusual foods would be of benefit to survival.

Thank you for this suggestion.

Lines 163-165: This is a conclusion and has no place in the Introduction.

We have removed this sentence from the introduction.

Methods should include the statistical methods that were employed including the statistical program that was utilised (later, the authors have mentioned, Wilcoxon, Friedman and a mixed generalized linear model). Without such information, subsequent researchers cannot repeat the work. The mark of a good Methods section is whether (or not) they are repeatable.

We have added a general statistical analysis section and we have added more clarifying details in the Results section to make sure the analyses could be replicated. We added, “All statistical analyses were conducted using SPSS v. 28.” We did not correct for multiple hypothesis testing with the planned contrasts because they were presented to describe and explain main effects, not to test independent hypotheses. We have now included a separate statistical analysis section.

I would also suggest a more explicit statement about the previous experience of these gorillas using a touch screen. Without such a statement, one might imagine that novelty could have contributed to the outcomes of the investigation.

We added, “all presented on a touchscreen” to the description of prior studies the gorillas had participated in.

Line 193: "to gorillas" Do the authors mean these gorillas or any gorillas? The authors need to be more precise in their language.

We changed to “subjects.”

Lines 201-202: "...tests were conducted sporadically ..." How many tests in total? Was the testing routine randomised both over time and across individuals? These questions are actually answered at Lines 225-. I suggest that the authors either include a cross-reference to this explanation or restructure the text to ensure that readers are not posing the same questions as me; this is just a matter of increasing the readability of the text.

Thank you for this suggestion. We reworded this part of the manuscript.

Lines 234-238: this is a repetition of the earlier part of the Methods (Lines 182-188). Restructure to avoid repetition.

We did repeat the time of testing because we found the information to be relevant both for the general procedure for the study and again for each specific phase of the experiments.

Lines 242-243: ", programmed in Real Basic," - repetition. I would also check whether the word “Basic” should be capitalized throughout as it is an acronym (Beginners All-purpose Symbolic Instruction Code).

Thank you for this. We checked and it is REALBasic. We deleted the second use to avoid redundancy.

Lines 313-314: I have decoded that SU=summer; FA=fall; and WI=winter. I am not sure that a reader should need to do this; I suggest that the authors explicitly state the season and year on the first mention.

We wrote out in full throughout.

Line 322 and on other occasions: the use of apostrophes to indicate plurals is grammatically incorrect.

All t’s

Yes, but we think this is an APA style convention for reporting multiple p and t values. We have removed the apostrophes throughout, regardless.

Line 326 (Legend to Fig 3): what do the error bars represent? This needs to be stated in the legend.

Thank you for this. We have added this information here

Lines 335-338: I presume that the CIs are 95%? This should have been stated in the Methods.

Thank you for noticing these omissions. We have added everywhere where we indicate confidence intervals.

Line 349: The acronym "AIC" is not explained.

We wrote it out in full here.

Lines 403-404: The authors report that preferences for beets and broccoli changed across seasons: how did the preference actually vary? And what reasons are you proposing (you need to provide some sort of explanation, or testable hypothesis)?

We did not have specific a priori hypotheses about why specific food preferences might change across seasons other than thinking that preferences for greens like lettuce might increase during winter when the gorillas could not forage in their outdoor habitats. We hypothesized that preferences would be stable across seasons so this was an unexpected finding.

Reviewer 3 Report

The authors present a novel research and I would like to see these findings published. There are, however, as series of issues that needs to be fixed before I can re-evaluate the paper. The main issue is the lack of description of the data analysis, meaning that I cannot assess whether the data were analysed in the correct way.

I present my comments in order:

  • I suggest to add more keywords (can add up to then so 4 is very limited) to make the paper easier to find.
  • line 44. I understand that the touch screen control for scent, but isn't scent of major importance for food selection? Not all the food items are selected based on colour vision. See for example: Melin, A.D., Nevo, O., Shirasu, M. et al. Fruit scent and observer colour vision shape food-selection strategies in wild capuchin monkeys.  Nat Commun 10, 2407 (2019). https://doi.org/10.1038/s41467-019-10250-9. Scent is only mentioned here, but I see this as a big limitation of this method and I suggest the authors to include a section in the discussion.
  • It might be just a style related to the discipline, but in my fields (behavioural ecology, ecology, conservation) the aim of the study goes in the final paragraph and not at the end of the first paragraph. So the structure of the introduction is now confusing to me. There is also a description of what you want to present in the middle of the introduction (in several parts), and that confusing.
  • You often talk about primates. Primates, however, are a diverse order, including various modes of colour vision (monochromacy, dichromacy, polymorphism with dichromacy and trichromacy, polymorphic trichromacy, and uniform trichromacy). Can this method be applied to all primates? Or should the authors consider just groups of primates when they discuss the applicability of the method? Also, can the method be applied to other species apart from primates (you report some other species but I suggest to include more species outside primates if possible)? 
  • line 253. This seems a big range considering that you say before that "A unique aspect of our program was that the researcher did not need to limit the foods presented to a predetermined number of expected foods". But it is clear that you needed to limit some of the food items. Why? And how did you control for this?
  • A figure (like a flowchart) with all the steps of the methods would be helpful to make it easier for the reader to understand all the steps and different phases of the data collection.
  • Lines 268-288. Some sentences of this paragraph seem results to be included in section 3.1. 
  • Line 281. Why did you present tomatoes and parsnips 3 times?
  • Line 307. What did you present more than 4 times? In section 2.3.3 it seems that you presented the items 2-3 times.
  • Line 308. Not sure about this result, there is no data analysis section so I do not know how the data were structured, what was the statistical unit, and why other tests that control for multiple individuals, e.g. generalised Linear Mixed Models, were not used.
  • In the results. Why do you expect differences between seasons? There is no introduction and it is not an aim. Also what are the seasons? No info in the methods? 
  • Line 313. Again a Generalised Linear Mixed Model maybe would be more suitable here. There should be a Data Analysis section explaining the choice of statistical tests and how they were performed.
  • Line 321. What do you mean by simple contrasts? Post-hoc tests following GLMMs? Did you correct the p-value for multiple hypotheses testing? 
  • Figure 3. There is not enough information to read the figure. What are those data? Estimated Model Means? Means from raw data? Confidence intervals? Standard errors?
  • Results 3.4 I am not sure you present here enough evidence of validation for the method. Why did you run a one-tailed one-sample t-test? Not sure how you structured the data. 

Author Response

The authors present a novel research and I would like to see these findings published. There are, however, as series of issues that needs to be fixed before I can re-evaluate the paper. The main issue is the lack of description of the data analysis, meaning that I cannot assess whether the data were analysed in the correct way.

Thank you for your support. We have attempted to be more explicit about the analyses, including adding a statistical analysis section.

I suggest to add more keywords (can add up to then so 4 is very limited) to make the paper easier to find.

We added three key words.

line 44. I understand that the touch screen control for scent, but isn't scent of major importance for food selection? Not all the food items are selected based on colour vision. See for example: Melin, A.D., Nevo, O., Shirasu, M. et al. Fruit scent and observer colour vision shape food-selection strategies in wild capuchin monkeys.  Nat Commun 10, 2407 (2019). https://doi.org/10.1038/s41467-019-10250-9. Scent is only mentioned here, but I see this as a big limitation of this method and I suggest the authors to include a section in the discussion.

We added a note upfront about the limitations of using touchscreen for this reason but we explained that we wished to control for these cues to food quality rather than type. We have expanded in the discussion on the limitation of removing a key source of discriminating information.

It might be just a style related to the discipline, but in my fields (behavioural ecology, ecology, conservation) the aim of the study goes in the final paragraph and not at the end of the first paragraph. So the structure of the introduction is now confusing to me. There is also a description of what you want to present in the middle of the introduction (in several parts), and that confusing.

We have restructured the introduction based on comments by all reviewers. In Psychology, we are trained to make the aim of the study clear in the first paragraph so that the reader is not wondering what the paper is really about as they read.

You often talk about primates. Primates, however, are a diverse order, including various modes of colour vision (monochromacy, dichromacy, polymorphism with dichromacy and trichromacy, polymorphic trichromacy, and uniform trichromacy). Can this method be applied to all primates? Or should the authors consider just groups of primates when they discuss the applicability of the method? Also, can the method be applied to other species apart from primates (you report some other species but I suggest to include more species outside primates if possible)?

We would suspect that foods can still be recognized if presented with black and white images. We have added some language to indicate the use of touchscreens may be appropriate outside of primates.

line 253. This seems a big range considering that you say before that "A unique aspect of our program was that the researcher did not need to limit the foods presented to a predetermined number of expected foods". But it is clear that you needed to limit some of the food items. Why? And how did you control for this?

We noted that we were limited by what foods were available in the gorillas’ daily morning food trays, but that there was a great deal of variety in what was presented both on a given day and across all test days. We were limited to the foods that were approved by the animal nutritionists and veterinary staff at the zoo. Some of the foods were rarely presented and some, like biscuits and tomatoes and Romaine lettuce were presented often. We did not control for this other than reducing our statistical analyses to foods that were presented more than four times across all the points of comparison (seasons or methods).

A figure (like a flowchart) with all the steps of the methods would be helpful to make it easier for the reader to understand all the steps and different phases of the data collection.

We have added a flowchart (Figure 1)

Lines 268-288. Some sentences of this paragraph seem results to be included in section 3.1.

The results from the other phase were used to create the stimuli for this phase. A description of the stimuli is necessary for the method of this particular phase.

Line 281. Why did you present tomatoes and parsnips 3 times?

The software program was designed for 20 trials (10 images from two categories) but not all gorillas had the appropriate number of food items that met our criterion for being considered preferred or less preferred so we adjusted the number of repetitions of those foods.

Line 307. What did you present more than 4 times? In section 2.3.3 it seems that you presented the items 2-3 times.

These sections are referring to two different phases/methods. Here, we are referring to the food preference tests not the test for discrimination of preferred versus less-preferred foods.

Line 308. Not sure about this result, there is no data analysis section so I do not know how the data were structured, what was the statistical unit, and why other tests that control for multiple individuals, e.g. generalised Linear Mixed Models, were not used.

We added more information about the data analyses in each results section and we added a general statistical analysis section.

In the results. Why do you expect differences between seasons? There is no introduction and it is not an aim. Also what are the seasons? No info in the methods?

We describe the seasons in the Method section for the touchscreen preference tests. We discussed in the introduction why we were interested in stability across seasons, but we have reminded readers of the relevance in the discussion.

Line 313. Again a Generalised Linear Mixed Model maybe would be more suitable here. There should be a Data Analysis section explaining the choice of statistical tests and how they were performed.

We have added this section. We did not conduct a GLMM here because we had only rankings of eight foods across two types of Method (the only effect being assessed). We were not assessing trial by trial data because we did not analyze the data for the many foods that were not presented consistently across sessions.

Line 321. What do you mean by simple contrasts? Post-hoc tests following GLMMs? Did you correct the p-value for multiple hypotheses testing?

We did not correct for multiple hypothesis testing with the planned contrasts because they were presented to describe and explain main effects, not to test independent hypotheses. In addition, we were hypothesizing no differences across seasons and within sessions, so it was more conservative not to correct. We have now included a separate statistical analysis section. We planned the contrasts when we conducted the GLMMs so they were not post-hoc tests.

Figure 3. There is not enough information to read the figure. What are those data? Estimated Model Means? Means from raw data? Confidence intervals? Standard errors?

We have provided more information about statistical analyses. These are estimated model means and standard errors, which we have now noted in the caption.

Results 3.4 I am not sure you present here enough evidence of validation for the method. Why did you run a one-tailed one-sample t-test? Not sure how you structured the data.

In this task, we were looking to see if there was a spontaneous preference for images of preferred over non-preferred food items so we compared the proportion of trials in which the preferred image was chosen to chance (.5 because there were only two options).

Thank you to all of the reviewers for their detailed and thoughtful comments,

Jennifer Vonk

Round 2

Reviewer 1 Report

I commend the authors for their revisions. I feel they have improved the paper a great deal in terms of readability, justification, and transparency. In deciding on the appropriateness for publication, I am primarily left with the debate of whether the methodological limitation of not pairing all foods with all other foods is enough of a limitation that the resulting data are not a useful contribution to the literature. After reflection, I feel these data do still still offer a contribution to the literature, and the authors have now been transparent enough in their explanations that I think readers can weigh the strengths and weaknesses of this methodology.

Here are my remaining comments/suggestions:

Line 169 requires more context. I suggest something like, “In this population, varied items were effective reinforcers for both basic tasks (bar presses) and….” (Once you say “bar press,” people think of animal studies, so I think it is important to make it clear that you are still talking about children here).

I suggest concluding the introduction with a paragraph that restates the different analyses you ran and your hypotheses for each.

The edits surrounding the selection of preferred vs non-preferred food photos has helped to clarify the purpose of this technique and improved clarity. However, I suggest removing the categorization of a “spontaneous” (line 26, 109, 484, and throughout paper) choice. Definitions of that word include “without external cause or stimulus” or “without premeditation” or “the result of a sudden internal urge.” The choice is most certainly based on past history (external factors such as reinforcement history), and you are unable to assess premeditation or urge in this case. It is best to give a more objective description of the scenario. For line 26, I would suggest, “Furthermore, gorillas selected images of preferred over less-preferred foods even in the absence of food item delivery…” or that you state “under extinction conditions.” In most contexts, you can just remove the word “spontaneous” and retain the meaning.

I suggest moving lines 326 – 329 earlier in this section to set up the difference in number of preferred foods from the beginning.

For readability, I suggest not using in-bracket citations as subjects/nouns in your sentences. For example changing, line 269 to “taken by researchers [31]” and line 474 to “six food items used in previous studies [10, 19].

Line 593. I suggest concluding this paragraph with a sentence or two that states some of the counterarguments the authors included in their author response. For example, “However, we do think that restricting our analyses to foods that were presented the most often (i.e., at least 37 times for the within session analyses) mitigates against this concern to some degree. Since each food item analyzed was paired with several different foods across all trials (i.e., foods were paired randomly rather than systematically), it is less likely that any particular food was consistently paired with higher or lower preference foods exclusively” and “We think that assessing the total proportion of trials chosen out of all of the trials offered accounts for the noise and variance in the particular food that the target food was paired with to the best of our ability given the manner in which the data was collected.” While some of this is referenced in the paragraph, a clear conclusion to this paragraph summarizing why these data have some merit despite this limitation would be helpful to readers.

Best wishes to the authors moving forward.

Author Response

I commend the authors for their revisions. I feel they have improved the paper a great deal in terms of readability, justification, and transparency. In deciding on the appropriateness for publication, I am primarily left with the debate of whether the methodological limitation of not pairing all foods with all other foods is enough of a limitation that the resulting data are not a useful contribution to the literature. After reflection, I feel these data do still still offer a contribution to the literature, and the authors have now been transparent enough in their explanations that I think readers can weigh the strengths and weaknesses of this methodology.

Thank you very much. We appreciate your careful consideration and hope that readers will also find the results of value despite their complicated nature.

Here are my remaining comments/suggestions:

Line 169 requires more context. I suggest something like, “In this population, varied items were effective reinforcers for both basic tasks (bar presses) and….” (Once you say “bar press,” people think of animal studies, so I think it is important to make it clear that you are still talking about children here).

We revised to, “In a human sample with developmental disorders, varied items, rather than a single item, served as effective reinforcers for basic tasks (i.e., a bar-pressing activity) as shown by higher rates or longer durations of responding [35, 36].”

I suggest concluding the introduction with a paragraph that restates the different analyses you ran and your hypotheses for each.

We added a section on tests and predictions without referring to the specific statistical approach, since that appears in its own section of the Methods.

The edits surrounding the selection of preferred vs non-preferred food photos has helped to clarify the purpose of this technique and improved clarity. However, I suggest removing the categorization of a “spontaneous” (line 26, 109, 484, and throughout paper) choice.  Definitions of that word include “without external cause or stimulus” or “without premeditation” or “the result of a sudden internal urge.” The choice is most certainly based on past history (external factors such as reinforcement history), and you are unable to assess premeditation or urge in this case. It is best to give a more objective description of the scenario. For line 26, I would suggest, “Furthermore, gorillas selected images of preferred over less-preferred foods even in the absence of food item delivery…” or that you state “under extinction conditions.” In most contexts, you can just remove the word “spontaneous” and retain the meaning.

We deleted the word “spontaneous” on line 26 and what was line 484. On line 109, we rephrased, “Preferences to touch images of preferred foods despite non-differential reinforcement would validate our classification of these foods into highly preferred and less preferred categories.” We wanted to emphasize the fact that selections of preferred foods were not differentially reinforced. We made five other similar changes.

I suggest moving lines 326 – 329 earlier in this section to set up the difference in number of preferred foods from the beginning.

Thank you for this excellent suggestion.

For readability, I suggest not using in-bracket citations as subjects/nouns in your sentences. For example changing, line 269 to “taken by researchers [31]” and line 474 to “six food items used in previous studies [10, 19].

This is also a good idea. Thank you.

Line 593. I suggest concluding this paragraph with a sentence or two that states some of the counterarguments the authors included in their author response. For example, “However, we do think that restricting our analyses to foods that were presented the most often (i.e., at least 37 times for the within session analyses) mitigates against this concern to some degree. Since each food item analyzed was paired with several different foods across all trials (i.e., foods were paired randomly rather than systematically), it is less likely that any particular food was consistently paired with higher or lower preference foods exclusively” and “We think that assessing the total proportion of trials chosen out of all of the trials offered accounts for the noise and variance in the particular food that the target food was paired with to the best of our ability given the manner in which the data was collected.” While some of this is referenced in the paragraph, a clear conclusion to this paragraph summarizing why these data have some merit despite this limitation would be helpful to readers.

We agree with this suggestion as well.

Best wishes to the authors moving forward.

Thank you so much for your very thoughtful reading of our work and all of your helpful suggestions!

Reviewer 3 Report

The manuscript is improved as is clear. I have no further comments to help improving it, apart from the following:

Line 148. Not sure what you mean by Pongo ns, maybe numerous species? That is not a correct way of reporting the name. It should be Pongo spp. if multiple species, Pongo sp. if only one species but you do not know the species name. 

Line 149. Ursus americanus

In the results, I suggest to use the same number of decimals for the similar values (e.g., p-values always 3, F and t values always 2, etc.). Also, I suggest adding zeros before the point (e.g, 0.16 not .16). That is just a suggested style.

Author Response

The manuscript is improved as is clear. I have no further comments to help improving it, apart from the following:

Thank you very much for your positive assessment!

Line 148. Not sure what you mean by Pongo ns, maybe numerous species? That is not a correct way of reporting the name. It should be Pongo spp. if multiple species, Pongo sp. if only one species but you do not know the species name. 

Thank you for this advice. We meant to indicate “not specified.” We have changed to your second suggestion.

Line 149. Ursus americanus

Thank you for this correction as well.

In the results, I suggest to use the same number of decimals for the similar values (e.g., p-values always 3, F and t values always 2, etc.). Also, I suggest adding zeros before the point (e.g, 0.16 not .16). That is just a suggested style.

Thank you for these suggestions. We have followed the first; however, we did not add leading 0s where the value could not be >1 as is the rule for APA. We are happy to add if the journal requires that we do so.

Thank you for your careful attention to our work and your helpful suggestions!